# Speeding up the Computation of the Transient Richards' Equation with AMGCL

**Robert Pinzinger** [1],[*] and **René Blankenburg** [2]

1    Institut für Grundwasserwirtschaft, Technische Universität Dresden, Bergstraße 66, 01069 Dresden, Germany
2    Ingenieurbüro für Grundwasser GmbH, Nonnenstraße 9, 04229 Leipzig, Germany;
     r.blankenburg@ibgw-leipzig.de
*    Correspondence: robert.pinzinger@tu-dresden.de

**Abstract:** The Richards'-equation is widely used for modeling complex soil water dynamics in the vadose zone. Usually, the Richards'-equation is simulated with the Finite Element Method, the Finite Difference Method, or the Finite Volume Method. In all three cases, huge systems of equations are to be solved, which is computationally expensive. By employing the free software library AMGCL, a reduction of the computational running time of up to 79% was achieved without losing accuracy. Seven models with different soils and geometries were tested, and the analysis of these tests showed, that AMGCL causes a speedup in all models with 20,000 or more nodes. However, the numerical overhead of AMGCL causes a slowdown in all models with 20,000 or fewer nodes.

**Keywords:** Richards'-equation; simulation; algebraic multigrid; preconditioner

## 1. Introduction

The vadose zone extends from the surface of the Earth down to the groundwater-table. This region is a habitat for bacteria and other microorganisms, some of which clean water while it travels through this zone. However, human activities in agriculture pollute the vadose zone and the groundwater with chlorine, nitrate [1], and hormones [2]. Industrial activities pollute the vadose zone and the groundwater with radioactive [3], petrochemical [4], and pharmaceutical [5] compounds. Some of these acts of pollution have the potential to change the properties of the vadose zone and the aquifer irreversibly [6] which in turn threatens the quality of groundwater in some regions permanently.

Land use can influence the amount of water in the vadose zone and the groundwater level [7]. The extraction of groundwater for irrigation can lead to land subsidence, especially in arid regions [8]. However, too much water in a slope may cause slope failure [9].

All this underlines the importance of being able to simulate the vadose zone. Accurate simulations of the vadose zone can aid in the decision on how to react to an incident of pollution. Also, changes in land use and their impact on the recharge of the aquifer can be simulated. Furthermore, the decision to evacuate areas under a slope can be supported with accurate simulations of the vadose zone.

The Richards'-equation is widely used for modeling complex soil water dynamics in the vadose zone [10]:

$$\frac{\partial \theta}{\partial t} = \sum_i \frac{\partial}{\partial i}\left[ K\left( \sum_j K_{i,j}^A \frac{\partial h}{\partial j} + K_{i,z}^A \right) \right] - S, \tag{1}$$

Here, $\theta$ represents the water content, $i, j \in \{x, z\}$ for a 2-dimensional formulation and $i, j \in \{x, y, z\}$ for a 3-dimensional consideration where $z$ is the vertical dimension, $K$ is the hydraulic conductivity, $K_{i,j}^A$ is the anisotropy tensor, $h$ is the pressure head, and $S$ is a sink term that models the in- and outflow over the system boundaries like evapotranspiration and precipitation. Thereby, Equation (1) formulates,

that the change in water content is caused by the seepage flow that is caused by pressure and gravity and by the in- and outflow of water over the system boundaries.

The aim of this study is to show that the numerical simulation of Equation (1) can be sped up by using the free software library AMGCL [11]. In Section 2, an overview on how equation systems that arise from numerical discretizations of Equation (1) are solved is given. In Section 3, AMGCL and PCSiWaPro [12] are introduced. The metrics that are used to compare the performance of the solver in PCSiWaPro and AMGCL are introduced in Section 4. In Section 5, models are presented that are used to make the comparison of the solver in PCSiWaPro and AMGCL. The results obtained from the simulations of these models are presented in Section 6. Section 7 finishes this study with a discussion and conclusion.

## 2. Solving Equation Systems that Arise from the Numerical Discretization of the Richards' Equation

There are several software packages available for the simulation of Equation (1). The most prominent are FEFLOW [13], FEMWATER [14], HydroGeoSphere [15], HYDRUS [16], PCSiWaPro, STOMP [17], SWMS [18], TOUGH3 [19], VS2D [20], and VSAFT2 [21].

Equation (1) is usually simulated with the Finite Element Method [22], the Finite Difference Method [22] or the Finite Volume Method [23].

Equation (1) is modelled with Finite Elements in FEFLOW, HydroGeoSphere, HYDRUS, PCSiWaPro, SWMS, and VSAFT2. The Finite Difference Method is used in STOMP, TOUGH3 and VS2D. The Finite Volume Method is considered in theoretical works [24,25], and also, there are some implementations of Equation (1) in the free numerical simulation toolbox OpenFOAM that discretize Equation (1) with the Finite Volume Method [26,27].

Whatever scheme is chosen to simulate Equation (1), the result is always an equation system of the form

$$Ax = b, \tag{2}$$

where $A$ is a matrix and $x$ and $b$ are vectors. In general, there are two ways to solve Equation (2) for $x$: The direct solution with the Gauss algorithm or "relatives" of it [22], or iterative methods [28].

The Gauss algorithm is computationally expensive which is why the Gauss algorithm is mainly used for small systems of equations. A "cousin" of the Gauss algorithm is the LU-decomposition [22], where one employs the Gauss algorithm to compute two matrices $L$ and $U$ where $L$ is a lower triangular matrix and $U$ is an upper triangular matrix, so that $LU = A$. A "cousin" of LU is the Cholesky factorization [22], which demands $A$ to be symmetric, where a lower triangular matrix $L$ is computed so that $LL^T = A$. The LU-decomposition or the Cholesky-decomposition are popular in situations, where one wants to solve an equation system as in Equation (2) with constant matrix $A$ and changing right hand sides $b$, because once the decompositions are computed, $Ax = b$ can be solved, in case of the LU-decomposition, as $LUx = b$, which in turn can be solved as $Ly = b$ with $y = Ux$. Both of these equation systems are triangular, and therefore these systems of equations can be solved very fast.

In the case of iterative procedures, one computes a sequence of approximate solutions $x_i$, so that the residual $r_i = b - Ax_i$ vanishes with growing $i$. There are several different schemes to compute this sequence. The most famous schemes are Gauss–Seidel [28], Krylow Subspace [28] algorithms, the Multigrid [28] method, and Stone's [29] method.

The basic idea of Gauss–Seidel is to decompose the matrix $A = D - E - F$ where $D$ is a diagonal matrix, $E$ is a lower triangular matrix and $F$ is an upper triangular matrix. From this, a recursion can be built: $x_{i+1} = (D - E)^{-1}Fx_i + (D - E)^{-1}b$. Instead of computing $y = (D - E)^{-1}v$ for a given vector $v = Fx_i$ or $v = b$, one solves $(D - E)y = v$ for $y$ which is computationally cheap since $(D - E)$ is a lower triangular matrix.

Krylow Subspace schemes employ the idea to compute the error $r_0 = b - Ax_0$, and to create a subspace of dimension $i$ from this error by defining $K_i = span\{r_0, Ar_0, \cdots, A^{i-1}r_0\}$. Then, one chooses $x_i \in x_0 + K_i$ so that $b - Ax_i$ is perpendicular on $L_i$, where $L_i$ is another subspace of dimension $i$. The

simplest choice is $L_i = K_i$ which is the starting point for the Conjugate Gradient algorithm [28]. In the Conjugate Gradient algorithm, the update of approximation $x_i$ to $x_{i+1}$ is made by employing a search direction $p_i$, that is computed with the vectors $r_i = b - Ax_i$ and $p_{i-1}$. The search directions $p_0, p_1, \ldots$ hold $(Ap_i)^T p_j = 0$ for $i \neq j$, which explains the term "conjugate" in Conjugate Gradient. ORTHOMIN [28] is a Krylow-type algorithm, where the search direction $p_i$ is computed with the vectors $r_i$ and $p_{i-m}, p_{i-m+1}, \ldots, p_{i-1}$, where the memory $m$ begins in the first iteration with $m = 1$ and grows with each iteration $i$ until a maximum $k$ is reached, where $m$ is restarted with $m = 1$. All Krylow Subspace algorithms aim at minimizing the norm of $r_i$, and the use of multiple past search directions as well as the reset of the number of past search directions every $k$ iterations help ORTHOMIN to not get stuck in local minima. Another Krylow-type algorithm is BICGSTAB [28] where the update of $x_i$ is made with a search direction $p_i$ and a vector $s_i$ that is a scaled version of the resulting residual which in turn is caused by the update of $x_i$ with $p_i$. Vector $s_i$ is scaled, so that the update of $x_i$ with $p_i$ and $s_i$ minimizes the norm of the residual $r_{i+1} = b - Ax_{i+1}$. This choice of $s_i$ stabilizes the convergence of BICGSTAB. Aside from the Conjugate Gradient algorithm, ORTHOMIN and BICGSTAB, there are many other Krylow Subspace schemes. A good overview over these schemes can be found in [28].

Krylow schemes can be sped up by preconditioning. In each iteration, the approximation error can be computed: $r_i = b - Ax_i$, then $x_i + A^{-1}r_i = x_i + A^{-1}Ax - A^{-1}Ax_i = x_i + x - x_i = x$. But since $A^{-1}$ is usually unknown and computationally expensive to compute, one searches for a matrix $M$ so that $M \approx A^{-1}$. A suitable matrix $M$ is called a preconditioner. The most famous preconditioner is ILU (Incomplete LU [28]), where one attempts to approximate $A$ by $LU \approx A$, where $L$ is a lower triangular matrix and $U$ is an upper triangular matrix. $L$ and $U$ are not computed so that $LU = A$ because this would be too expensive computationally. A vector $v$ is preconditioned with ILU by solving $LUy = v$ for $y$, which is computationally cheap since $L$ and $U$ are both triangular matrices. However, this scheme cannot be parallelized, because in order to solve a triangular equation system, one has to solve the rows consecutively. Another popular preconditioner is SPAI (sparse approximate inverse [30]). Here, one attempts to compute a matrix $M$, so that the sum over the squares of the entries of $I - MA$ is minimal, where $I$ is the identity matrix. The advantage of SPAI is, that SPAI is constructed in a way that $M$ can be computed massively parallel. There also exists polynomial preconditioning [28]: consider $\widetilde{A} = \omega A = I - B$, where $\omega$ is chosen, so that the largest absolute eigenvalue of $B$ is smaller than 1, then the following relationship holds: $\widetilde{A}^{-1} = \sum_k B^k$. This preconditioner is computationally very expensive since it demands many matrix-matrix-multiplications, but it can be massively parallelized. Also, this preconditioner demands enormous amounts of storage. Usually, the matrices involved with Finite Element, Finite Difference, or Finite Volume models are sparse, which means, that each row of the matrices has only a few entries which need to be kept in storage. If one multiplies matrices with each other, this sparsity is lost in general, which results in huge demands for storage.

Another approach to solve large equation systems is the Multigrid for which there are two variations: The geometric Multigrid and the algebraic Multigrid. In the case of the geometric Multigrid [28], the discretization of the problem underlying Equation (2) is mapped to a sequence of consecutive coarser discretizations, and for each level of discretization, an equation system is created. Thereby, the equation system gets smaller with each level of discretization. On a given level, the current approximation error is reduced by a simple iterative scheme. This reduction of the approximation error is called relaxation. Simple algorithms are preferred for the relaxation, because one does not want to compute the exact solution, but rather a good approximation which can be computed fast. Algorithms like Gauss–Seidel or SPAI0 [31] are popular algorithms for relaxation. SPAI0 defines a smoother $x_{i+1} = x_i - M(Ax_i - b)$, where the matrix $M$ is a diagonal matrix with $M_{k,k} = \frac{A_{k,k}}{\sum_j A_{k,j}^2}$. Due to its simple structure, $M$ can be computed in parallel. Also, the smoother SPAI0 can be highly parallelized, since it consists only of matrix-vector products and vector subtractions. On a given level, after the relaxation, the current approximation is either mapped to a coarser level or to a finer level. The mapping of the current approximation from a coarse- to a fine level is called prolongation, the mapping from a fine level to a coarse level is called restriction. There are different schemes like the V-cycle [28], where one

starts on the finest level, relaxes and restricts until the coarsest level is reached, and then relaxes and prolongs until the finest level is reached, where the sequence of restrictions and prolongations forms a "V". In the W-cycle [28], one arranges the restrictions and prolongations, so that they form a "W". Since the dimensionality of the equation systems sinks with the coarseness of a level, the operations on the coarse levels are computationally very cheap. To sum up the idea of the geometric Multigrid, one maps a discretized geometry to a sequence of discretized geometries with increasing coarseness. For each of these grids, an equation system is formulated and the solution on each grid is approximated and mapped to the next grid. The problem with the geometric Multigrid is, however, that it shows difficulties with anisotropic coefficients of the underlying partial differential equation [28]. These problems arise from the fact, that if the coefficients of the underlying partial differential equation are anisotropic, then errors are automatically introduced into the solution with the step from the fine grid to the coarse grid and vice versa. In the case of Equation (1), the coefficients are anisotropic, because each point in space has a pressure head which defines the hydraulic conductivity at that point. Therefore, if the pressure heads are anisotropic, then the hydraulic conductivity in Equation (1) is anisotropic. To illustrate the problem, consider the simple model of two nodes, where both nodes have a unique pressure head. If one would coarsen this simple model, one would restrict both nodes to one node. The node on the coarser level, however, can only reflect the hydraulic conductivity of at most one node from the finer level. Therefore, geometric Multigrid solvers perform poorly on discretizations of Equation (1).

In the case of the algebraic Multigrid, one does not bother with the geometry of the discretization, but rather one creates from the original equation system a sequence of equation systems with decreasing sizes [28]. There are several schemes to compute the smaller equation system from a given equation system, the most famous schemes are the ones by Ruge and Stüben [32] and Smoothed Aggregation [33]. The basic idea of Ruge and Stüben is, to interpret a matrix $A$ as a graph, where the entry $A_{k,j}$ tells, how strongly the row $k$ is connected to the row $j$. Then, one uses this strength of connection to decide, which rows need to be represented in the next coarser level, the $C$ rows, and which rows need not to be represented, the $F$ rows. In the Ruge and Stüben scheme, $k$ is strongly negative coupled to $j$ if $-A_{k,j} \geq \varepsilon \max_{A_{k,l}<0} |A_{k,l}|$ for some $0 < \varepsilon < 1$. This strongly negative coupling is then used to define the set $S_k = \left\{ j : A_{k,j} \neq 0, k \text{ strongly negative coupled to } j \right\}$. The transpose of this set is $S_k^T = \left\{ j : k \in S_j \right\}$. Therefore, $S_k^T$ contains all $j$ that are strongly negative coupled to $k$. Once, the sets $S_k^T$ are computed for each row, one iteratively selects one $k$, puts this row into the $C$ set, and the rows in $S_k^T$ into the $F$ set. Since this selection has to be performed iteratively, it cannot be parallelized. However, the information from the rows in the $F$ set must not vanish, therefore the information from these rows is interpolated into the rows from the $C$ set. Thereby, the problem with anisotropic coefficients in the geometric Multigrid is circumvented. In the Smoothed Aggregation algorithm, for each row $k$, the neighborhood is defined by $N_k(\varepsilon) = \left\{ j : |A_{k,j}| \geq \varepsilon \sqrt{A_{k,k}A_{j,j}} \right\}$. For a given $k$, $k$ is put into the $C$ set, and the $j$ in $N_k(\varepsilon)$ are put into the $F$ set. As with the Ruge and Stüben algorithm, the information from the rows in the $F$ set is interpolated into the rows from the $C$ set. In both algorithms, the solution is computed as with the geometric Multigrid once the equation systems are created.

The last approach to solve large equation systems that shall be considered in this paper is Stone's Method. The core of Stone's Method is to decompose $A = S - T$. Then, one can formulate the iterative scheme $Sx_{i+1} = Tx_i + b$. Now, one has to solve this equation system for $x_{i+1}$ which is why one chooses $S$, so that the decomposition $S = LU$, where $L$ is a lower triangular matrix and $U$ is an upper triangular matrix, is easy to compute.

FEFLOW offers direct Gaussian-type solvers, Krylow Subspace schemes and geometric and algebraic Multigrid. The Krylow schemes can be preconditioned with ILU. FEMWATER has Gauss–Seidel and preconditioned Conjugate Gradient solvers. The preconditioners are ILU and polynomial. HydroGeoSphere uses ORTHOMIN with ILU as preconditioner. HYDRUS, PCSiWaPro, STOMP and SWMS employ the Gauss-algorithm and preconditioned Conjugate Gradient with ILU

as preconditioner. TOUGH3 offers the Gauss-algorithm and ILU-preconditioned Krylow Subspace schemes. VS2D uses Stone's method as solver, and VSAFT2 has the ILU-preconditioned Conjugate Gradient as solver.

## 3. AMGCL and PCSiWaPro

As one can see, most software packages that solve the Richards'-equation with Krylow-type algorithms employ the ILU preconditioner. As shown above, ILU suffers from two aspects: On the one hand, ILU itself can in general only be a poor approximation to the real inverse, and on the other hand, ILU cannot be parallelized which makes preconditioning with ILU slow. Hence, there is a need for a highly parallelizable and yet accurate preconditioner for Krylow-type solvers of equation systems arising from discretizations of Equation (1). Because if one had a highly parallelizable and yet accurate preconditioner, then this preconditioner would speed up the solution of equation systems arising from discretizations of Equation (1). This speedup can then be translated into larger models that simulate the system in question in more detail, or one could use this additional time for simulating multiple scenarios.

Demidov published AMGCL, a C++-library with several Krylow-type solvers for which there are several algebraic Multigrid preconditioners available. AMGCL supports parallelization via OpenMP [34], CUDA [35] and MPI [36]. In the standard settings, AMGCL performs preconditioning with an algebraic Multigrid and employs the V-cycle, and the equation system is coarsened until the coarsest level has at most 3000 nodes. On the coarsest level, a LU-solver is employed. According to the standard settings of AMGCL, on each level two iterations of relaxation are performed. AMGCL offers algebraic Multigrid preconditioning according to the Ruge and Stüben scheme and Smoothed Aggregation. For the relaxation, there are Gauss–Seidel-, ILU- and SPAI-smoothers.

The development of PCSiWaPro began in the early 2000s when a group of scientists at the Technical University of Dresden took the numerical simulation code of SWMS_2D and extended this code by coupling the numerical kernel of SWMS_2D with a GUI that allows the user to create, modify and save 2-dimensional Finite Element models of the Richards' equation. Also, PCSiWaPro extends SWMS_2D by offering a database that contains the properties of various soils according to DIN 4220 [37] which is a standard for the designation, classification and deduction of soil parameters in Germany. The parameters of the soils can also be set manually. In order to extend the modelling capabilities of PCSiWaPro, the numerical kernel of SWMS_3D [38] was added in 2019. Furthermore, the extension "Weather Generator" allows to create atmospheric boundary conditions from measurement data. PCSiWaPro uses the van Genuchten (which shall be abbreviated by VG) model [39,40] to describe the retention curve and the unsaturated hydraulic conductivity function of the soils in the model. As stated above, PCSiWaPro uses a Gaussian-type solver (for equation systems with less than 500 dimensions) and the ILU-preconditioned Conjugate Gradient (for equation systems with 500 or more dimensions) to solve equation systems that arise from the numerical approximation of Equation (1). A relative and an absolute tolerance of $1e^{-6}$ and a maximum of 1000 iterations are used as criteria for convergence and divergence. PCSiWaPro supports varying time step sizes. If the computation of a time step costs less than 4 calls of the solver, then the time step size is increased by a user defined factor. If the computation of a time step costs more than 6 calls of the solver, then the time step size is reduced by a user defined factor.

In order to investigate whether AMGCL can speed up the simulation of Equation (1), AMGCL was integrated into PCSiWaPro. The AMGCL solver was set up to employ BICGSTAB with an algebraic Multigrid-preconditioner which is based on Smoothed Aggregation where SPAI0 is used for relaxation.

BICGSTAB was chosen, due to its stabilized convergence behavior which makes the solver more robust. Initial tests revealed, that BICGSTAB preconditioned with algebraic Multigrid based on Smoothed Aggregation computes faster than BICGSTAB preconditioned with algebraic Multigrid based on the algorithm by Ruge and Stüben, therefore algebraic Multgrid with Smoothed Aggregation was chosen as preconditioner. SPAI0 was chosen for the relaxation, because SPAI0 was designed with

the intention of massive parallelization, whereas, for example, the Gauss–Seidel scheme or ILU cannot be parallelized really well since triangular equation systems have to be solved.

Since the performance of AMGCL is to be compared with the ILU-preconditioned Conjugate Gradient solver, the absolute and relative tolerance of 1e-6 and a maximum of 1000 iterations were coded into AMGCL to have a fair comparison.

PCSiWaPro with the AMGCL-solver with BICGSTAB as solver and algebraic Multigrid preconditioning based on the Smoothed Aggregation scheme with SPAI0 relaxation will be from now on referenced as PCSiWaPro AMGCL, whereas PCSiWaPro with the ILU-preconditioned Conjugate Gradient will be referenced as PCSiWaPro Original.

## 4. Method

PCSiWaPro AMGCL and PCSiWaPro Original were compared on 7 synthetic models which will be described in the next section. These models were tuned with computations run in PCSiWaPro Original. The number of nodes in the models ranges from 7000 nodes to 2,400,000 nodes. Two models consider simple rectangular shapes, one model considers a rectangular cuboid, whereas the other four models consider more complex geometries. Four models consist of only one type of soil, the other three models each have at least two different kinds of soil.

All computations for the 6 2-dimensional models were run on a desktop PC running Windows 8 with an Intel Core i7-6700 CPU and 8 GB of RAM. During the computations, all 8 cores that are available on the i7-6700 CPU were used since PCSiWaPro and AMGCL both support parallelization with OpenMP. The 3-dimensional model was simulated on a computer running Windows Server 2012 r2 with an Intel Core i7-6800K CPU and 32 GB of RAM, and all 6 cores were used during the computations.

For the comparison of PCSiWaPro AMGCL with PCSiWaPro Original, three metrics were considered: the computational running time of the simulations which was measured in seconds, the $R^2$-value between the pressure heads computed with PCSiWaPro Original and PCSiWaPro AMGCL, and the relative cumulative mass balance error (which shall abbreviated with RCMBE) computed with these two programs.

The $R^2$-value between the pressure heads computed with PCSiWaPro Original and PCSiWaPro AMGCL was chosen as a metric, because PCSiWaPro solves Equation (1) for the pressured heads.

$$R^2(x_i, y_i) = 1 - \frac{\sum_j \left(x_i^j - y_i^j\right)^2}{\sum_j \left(x_i^j - \mu_i\right)^2}, \tag{3}$$

The $R^2$-value was computed with Equation (3), where $x_i$ is the vector of the pressure heads computed with PCSiWaPro Original at time point $t_i$, and $x_i^j$ is the $j$-th entry of this vector. $\mu_i$ is the mean of the values in $x_i$. $y_i$ is the vector of the pressure heads computed with PCSiWaPro AMGCL at time point $t_i$, and $y_i^j$ is the $j$-th entry of this vector. The nominator in Equation (3) is the squared error metric that computes the squared distance between the vectors $x_i$ and $y_i$. This distance is normalized by the denominator. The normalization allows the $R^2$-value of different time steps or models to be compared. The best possible $R^2$-value is 1 when $x_i$ and $y_i$ are identical. The smaller the $R^2$-value, the greater the distance between the vectors $x_i$ and $y_i$. The $R^2$-value was computed for every time step, and thereby, for each model, a time series of $R^2$-values was computed. The $R^2$-values were rounded to 4 digits after the decimal point.

$$RCMBE(t_i) = \frac{\left|V_{t_i} - V_{t_0} + \int_{t_0}^{t_i} RWU(t)dt - \int_{t_0}^{t_i} \sum_{j \in \Gamma} Q_j(t)dt\right|}{\max\left(\sum_e \left|V_{t_i}^e - V_{t_0}^e\right|, \int_{t_0}^{t_i}\left(RWU(t) + \sum_{j \in \Gamma}|Q_j(t)|\right)dt\right)} \tag{4}$$

The RCMBE was computed with Equation (4). Here, $V_{t_0}$ is the amount of water in the model at time point $t_0$. $V_{t_i}$ is the amount of water in the model at time point $t_i$. $RWU(t)$ is the root water uptake

at time point $t$. $\Gamma$ is the set of boundary nodes of the model. $Q_j(t)$ is the flow over the boundary in boundary node $j$ at time point $t$. $V_{t_0}^e$ is the amount of water in element $e$ at time point $t_0$, $V_{t_i}^e$ is the amount of water in element $e$ at time point $t_i$. Therefore, the nominator in Equation (4) computes the absolute cumulative mass balance error at time point $t_i$ by considering the difference in water mass between the first time point $t_0$ and the current time point $t_i$, the root water uptake between the first time point $t_0$ and the current time point $t_i$, and the in- and outflow of water over the system boundaries between the first time point $t_0$ and the current time point $t_i$. The denominator compares two terms and chooses the bigger one. The first term is the sum of the absolute changes in water mass in each element between the first time point $t_0$ and the current time point $t_i$. The second term is the integral over the root water uptake and the sum of the absolute values of the mass transfer over system boundaries in the boundary nodes. The RCMBE is a proxy to how well the numerical approximation works, because if there were no numerical approximation and no rounding errors, the nominator in Equation (4) would be 0, because the change in water mass between the first and the current time point would be explained by the amount of water removed by the plants and the amount of water that left and entered the system over the boundaries. If the nominator in Equation (4) is not equal to 0, then the numerical approximation and rounding errors created or destroyed water artificially. The code of PCSiWaPro Original was used during the tuning of the models that will be presented in the next section. One aim of the tuning was, to keep the RCMBE below 1% for all time points in the simulation period.

## 5. Description of Synthetic Models

Figure 1 gives a first overview over the 2-dimensional models used in the comparison. The properties of their soils can be read in Table 1. Figure 2 gives an overview over Model 7.

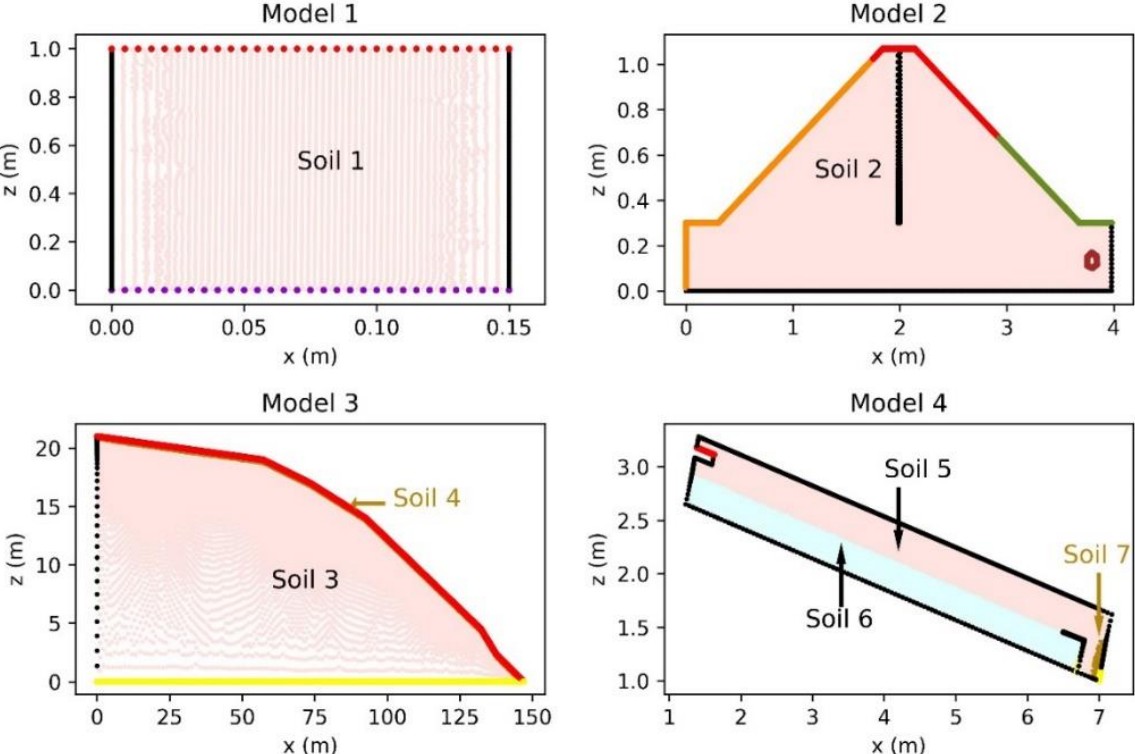

**Figure 1.** *Cont.*

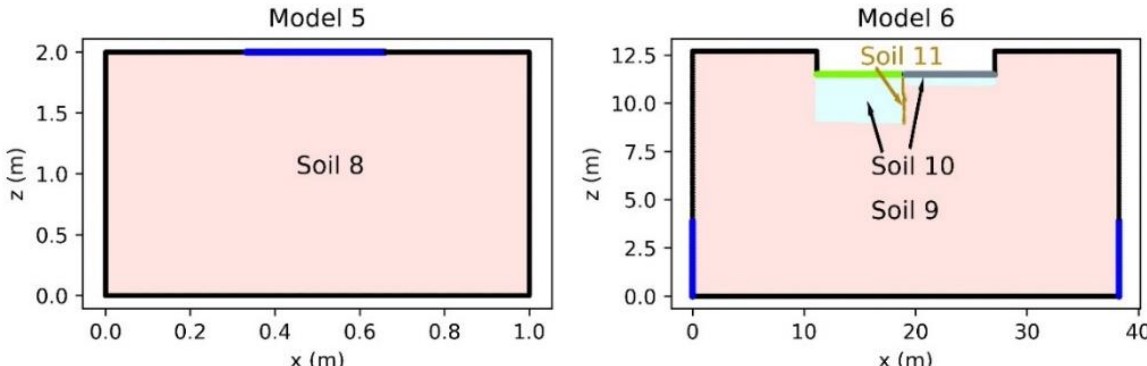

**Figure 1.** An overview over the shape of models 1 to 6, their sizes and the layout of the soil layers. The colors mistyrose, lightcyan, and darkgoldenrod are reserved for soils. Black depicts a no-flow boundary condition, blue depicts a constant pressure head boundary condition. Red, green, and grey depict a time dependent infiltration rate, darkviolet and brown depict a system dependent boundary condition: As long as the nodes are not saturated, darkviolet and brown nodes depict a no-flow boundary condition, and once the nodes are saturated, the boundary condition switches to a boundary condition with a constant pressure head of 0 m. Orange depicts a time dependent pressure head, olive depicts a seepage face, yellow depicts a constant pressure head and only outflow is allowed in yellow nodes.

**Table 1.** Overview over the parameters of the soils in Figures 1 and 2.

| Soil | Porosity | Permeability | Residual Air Content | Residual Water Content | VG $\alpha$ | VG n | VG m | VG $\lambda$ |
|------|----------|--------------|----------------------|------------------------|-------------|------|------|--------------|
| - | % | m/s | % | % | 1/m | - | - | - |
| 1 | 39 | $5 \times 10^{-5}$ | 0 | 5 | 3 | 2.2 | 0.545 | 0.5 |
| 2 | 40 | $1.67 \times 10^{-5}$ | 0 | 2 | 4 | 1.9 | 0.47 | 0.5 |
| 3 | 30 | $1 \times 10^{-3}$ | 0 | 1 | 55 | 2 | 0.5 | 0.5 |
| 4 | 40 | $1 \times 10^{-5}$ | 0 | 5 | 1 | 1.6 | 0.33 | 0.5 |
| 5 | 40 | $1.06 \times 10^{-4}$ | 0 | 4 | 4.5 | 4 | 0.75 | 0.5 |
| 6 | 30 | $1.83 \times 10^{-2}$ | 0 | 1 | 100 | 3 | 0.67 | 0.5 |
| 7 | 30 | $1.06 \times 10^{-4}$ | 0 | 4 | 4.5 | 4 | 0.75 | 0.5 |
| 8 | 43 | $2.89 \times 10^{-6}$ | 0 | 7.8 | 3.6 | 1.56 | 0.36 | 0.5 |
| 9 | 41 | $2 \times 10^{-5}$ | 0 | 5.7 | 12.4 | 2.28 | 0.561 | 0.5 |
| 10 | 36 | $1.65 \times 10^{-3}$ | 0 | 1 | 35 | 3 | 0.666 | 0.5 |
| 11 | 43 | $1.16 \times 10^{-8}$ | 0 | 7.8 | 3.6 | 1.56 | 0.641 | 0.5 |
| 12 | 43 | $2.89 \times 10^{-6}$ | 0 | 7.8 | 3.6 | 1.56 | 0.36 | 0.5 |

*5.1. Model 1*

Model 1 is a column of 1 m in height and 0.15 m in width. It is based on a setup of an infiltration experiment with polluted water. The upper boundary condition is described by a time-dependent infiltration rate (depicted in red in Model 1 in Figure 1) whereas the lower boundary condition is system dependent (depicted in darkviolet in Model 1 in Figure 1): while the lower boundary is unsaturated, it is handled as a no-flow boundary condition and the pressure head is calculated. As soon as the lower boundary is saturated, the boundary condition switches to a boundary condition of the first type with a prescribed pressure head of zero, and thus, the outflow is calculated. The model is discretized with 7178 nodes. The parameters of Soil 1 from Table 1 were used to describe the hydraulic properties of the soil (depicted in mistyrose in Model 1 in Figure 1). The black nodes in Model 1 in Figure 1 depict a no-flow boundary condition.

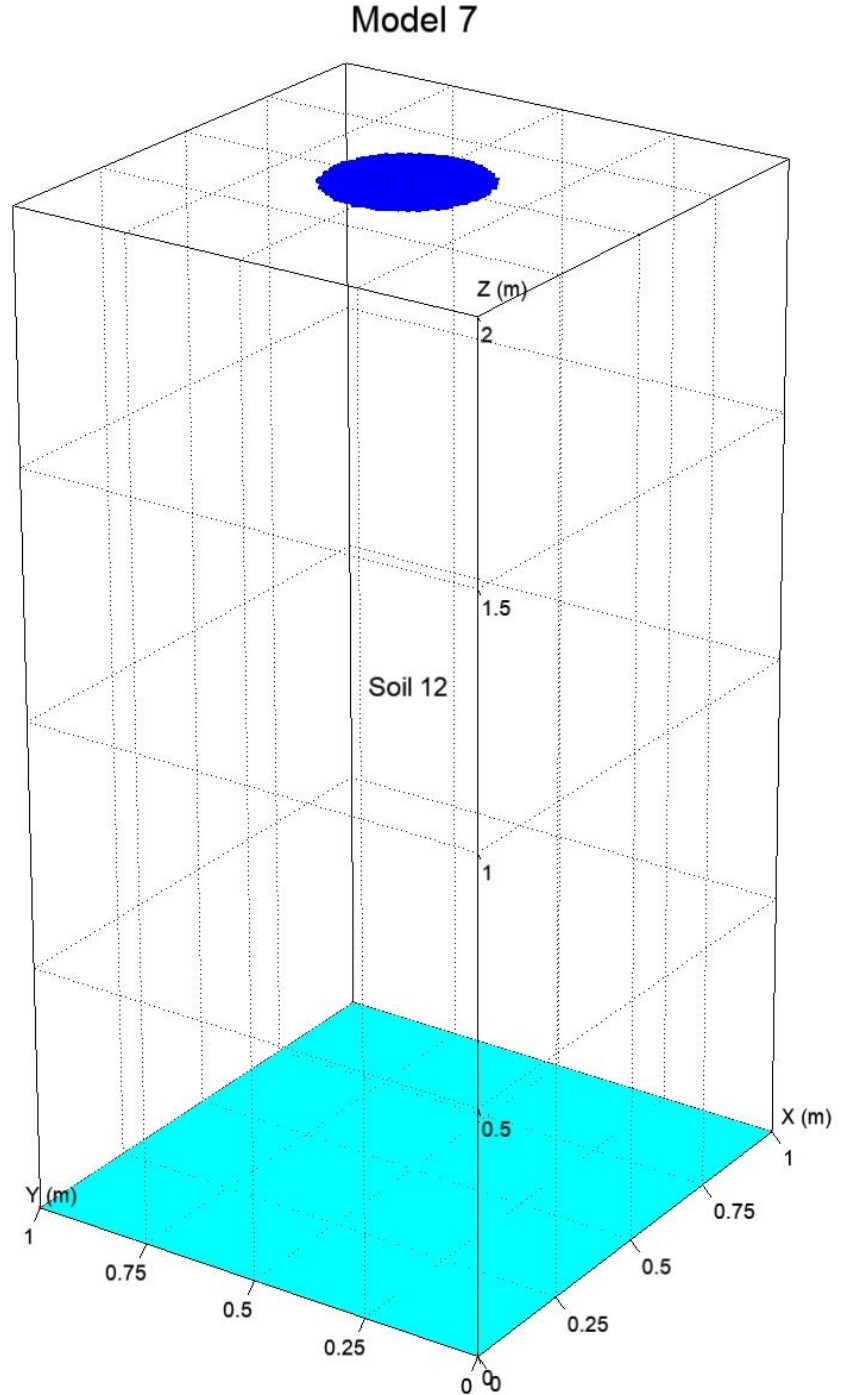

**Figure 2.** Overview over Model 7. Blue depicts a constant pressure head boundary condition, cyan depicts a free drainage boundary condition. The rest of the surface of the cuboid has a no-flow boundary condition which is not depicted.

### 5.2. Model 2

Model 2 is based on a physical experiment setup [41] and describes a small-scale levee of a homogenous soil. The foot-width of the model-scale levee is 3.38 m, the slope angle of 1:2 according to a height of 0.77 m. At the levee-crest, a rubber wall is installed having an anchoring depth of 100% of the dam height. Below the levee, a berm with a height of 0.3 m was added at both, the water and the valley side of the dam, to allow for exchange of the inflowing water with the groundwater. On the downstream face, a drainage tube was implemented to effectively discharge excess water (depicted in

brown in Model 2 in Figure 1). The model is discretized into 11,896 nodes. The parameters from Soil 2 in Table 1 were used for the hydraulic properties of the homogenous soil layer, the soil is depicted in mistyrose in Model 2 in Figure 1. The black nodes in Model 2 in Figure 1 depict a no-flow boundary condition. A flood is simulated by a time dependent pressure head depicted in orange nodes in Model 2 in Figure 1. Precipitation is modeled by a time dependent infiltration rate depicted in red nodes in Model 2 in Figure 1. The olive nodes in Model 2 in Figure 1 model a seepage face where the flow is 0 and the pressure head is computed.

### 5.3. Model 3

Model 3 is a landfill site of 20 m height and 150 m width which is constructed with a surface sealing system using the principle of a capillary barrier. It consists of a coarse layer as capillary block (filling material, Soil 3 in Table 1, depicted in mistyrose in Model 3 in Figure 1) and a fine layer as cover (Soil 4 in Table 1, depicted in brown in Model 3 in Figure 1). The aim of the model is to prove the functionality of the sealing system under different precipitation scenarios and thicknesses of the cover layer. Therefore, the upper boundary condition (depicted in red in Model 3 in Figure 1) infiltrates water with varying intensity directly into the cover layer whose thickness ranges from 0.1 m to 0.3 m. It is modelled with 53,795 nodes. The yellow nodes in Model 3 in Figure 1 depict a constant pressure boundary condition where only outflow is allowed, whereas the black nodes in Model 3 in Figure 1 model a no-flow boundary condition.

### 5.4. Model 4

Model 4 is based on a physical experiment of a tipping trough. Its purpose is to simulate the proper functionality of a capillary barrier under different inflow scenarios, slope angles, and hydraulic properties of the built-in soil materials. The model used for the tests in this paper has a height of 0.6 m, a length of 6 m and a slope angle of 0.277 rad. Three different soil types are used: a cover layer (capillary layer, Soil 5 in Table 1, depicted in mistyrose in Model 4 in Figure 1), a bottom layer (capillary block, Soil 6 in Table 1, depicted in lightcyan in Model 4 in Figure 1) and a layer of filter sand (Soil 7 in Table 1, depicted in brown in Model 4 in Figure 1). On the upper left side, a water inlet with a time-dependent influx (depicted in red nodes in Model 4 in Figure 1) is simulated while balancing the outflow in the capillary block layer and in the filter sand (depicted in yellow nodes in Model 4 in Figure 1). The black nodes in Model 4 in Figure 1 portray a no-flow boundary condition. To account for the large gradient expected during the simulation run, the model is relatively fine discretized into 21,221 nodes.

### 5.5. Model 5

Model 5 is a column of 2 m height and 1 m in width. The properties of the soil are described as Soil 8 in Table 1, depicted in mistyrose in Model 5 in Figure 1. The model is infiltrated by a point source on the top of the column, depicted in blue in Model 5 in Figure 1. Aside from this, there is a no-flow boundary condition along the boundary of the model, depicted in black nodes in Model 5 in Figure 1. It is a synthetic, medium-scale model whose purpose is mainly to check the scalability of the numerical approach. It consists of 501,501 nodes.

### 5.6. Model 6

Model 6 simulates a mini sewage plant with two neighboring infiltration basins. One basin is for the infiltration of treated sewage water, the other is for the infiltration of rain. The dimensions of the model are 38 m in width and 12 m in height. The infiltration basins both have a width of 6.5 m. Under the infiltration basin for the rain water, there is a layer of gravel of 1.5 m thickness which is modelled with Soil 10 in Table 1, depicted in lightcyan in Model 6 in Figure 1. Under the infiltration basin for the treated sewage water, there is a layer of gravel with a thickness of 0.5 m which is modelled with Soil 10 from Table 1, depicted in lightcyan in Model 6 in Figure 1. Between these two basins, a barrier

(Soil 11 from Table 1, depicted in brown in Model 6 in Figure 1) is installed to prevent immediate mixing of the infiltrated waters. The rest of the model is modelled with the sandy Soil 9 from Table 1, depicted in mistyrose in Model 6 in Figure 1. The model is infiltrated through both infiltration basins with individual infiltration rates and following individual time patterns, depicted in green (for the infiltration basin for the rain) and grey (for the infiltration basin for the treated sewage water) in Model 6 in Figure 1. On the vertical boundaries of the model, there are two constant pressure head boundary conditions to simulate steady-state groundwater, depicted in blue in Model 6 in Figure 1. Aside from these two boundary conditions and the infiltration basins, there is a no-flow boundary condition along the boundary of the model, depicted in black in Model 6 in Figure 1. The model consists of 54,763 nodes.

### 5.7. Model 7

Model 7 is a rectangular cuboid of 2 m height, 1 m width, and 1 m depth. The properties of the soil are described as Soil 12 in Table 1. The model is infiltrated by a circular source with a diameter of 0.33 m on the top of the cuboid (portrayed in blue in Figure 2). On the bottom, there is a free drainage boundary condition (depicted in cyan in Figure 2), and aside from this boundary condition and the circular source, there is a no-flow boundary condition along the boundary of the model. It is a synthetic, large-scale model whose purpose is mainly to check the scalability of the numerical approach. It consists of 2,377,026 nodes.

## 6. Results and Discussion

### 6.1. Comparison of the Computational Running Time

In the comparison in Table 2, the median speedup of PCSiWaPro AMGCL over PCSiWaPro Original is 16.8%. However, there is a huge variance in the speedup. Figure 3 shows the relationship between the speedup achieved with PCSiWaPro AMGCL over PCSiWaPro Original in relation to the number of nodes in the model. One clearly sees, that PCSiWaPro AMGCL outperforms PCSiWaPro Original in models with 20,000 or more nodes in terms of speed. This can be explained in the following way: The creation of the algebraic Multigrid costs computational time. Also, operations on the grids cost computational time, but these costs are negligible for very coarse grids if the original grid is very fine. In AMGCL, the coarsest grid has at most 3000 nodes, therefore, if the original equation system has 10,000 nodes, operations on the coarsest grid cost one third of the computational cost of the same operation on the finest grid. If the original equation system has 300,000 nodes, the operations on the coarsest grid will cost one percent of the computational cost of the same operation on the finest grid. Therefore, with growing size of the problem, the cost for operating on the coarser grids becomes negligible. And this explains why the speedup of PCSiWaPro AMGCL over PCSiWaPro Original grows with the number of nodes in the model. The last two columns in Table 2 support this reasoning for models 1 to 3 and models 5 and 6: in the simulation runs of these models, PCSiWaPro Original and PCSiWaPro AMGCL called the solver approximately equally often, however, there is the huge difference in computational running time documented in the first two columns of Table 2. Since the code of PCSiWaPro AMGCL is identical with the code of PCSiWaPro Original except for the solver, this difference in computational running time can only be explained by the computational running time that is spent with the respective solvers. Since BICGSTAB and the Conjugate Gradient algorithm are similar in their computational complexity, the difference in computational running time can only be attributed to the computational complexity of the preconditioner and the quality with which the preconditioner approximates the inverse matrix.

**Table 2.** Overview over the computational running time and the number of calls of the solver of the simulations of models 1 to 7 with PCSiWaPro Original and PCSiWaPro AMGCL.

| Model | Computational Running Time of PCSiWaPro Original | Computational Running Time of PCSiWaPro AMGCL | Speedup of PCSiWaPro AMGCL in Relation to PCSiWaPro Original | Number of Calls of the Solver in PCSiWaPro Original | Number of Calls of the Solver in PCSiWaPro AMGCL |
|---|---|---|---|---|---|
| - | s | s | % | - | - |
| 1 | 168 | 258 | −53.6 | 26,103 | 26,082 |
| 2 | 4 | 8 | −100 | 370 | 370 |
| 3 | 1751 | 784 | 55.2 | 10,824 | 9459 |
| 4 | 5698 | 3696 | 35.1 | 239,754 | 126,298 |
| 5 | 1856 | 382 | 79.4 | 464 | 490 |
| 6 | 9318 | 7755 | 16.8 | 118,526 | 113,476 |
| 7 | 5730 | 4860 | 15.2 | 733 | 836 |

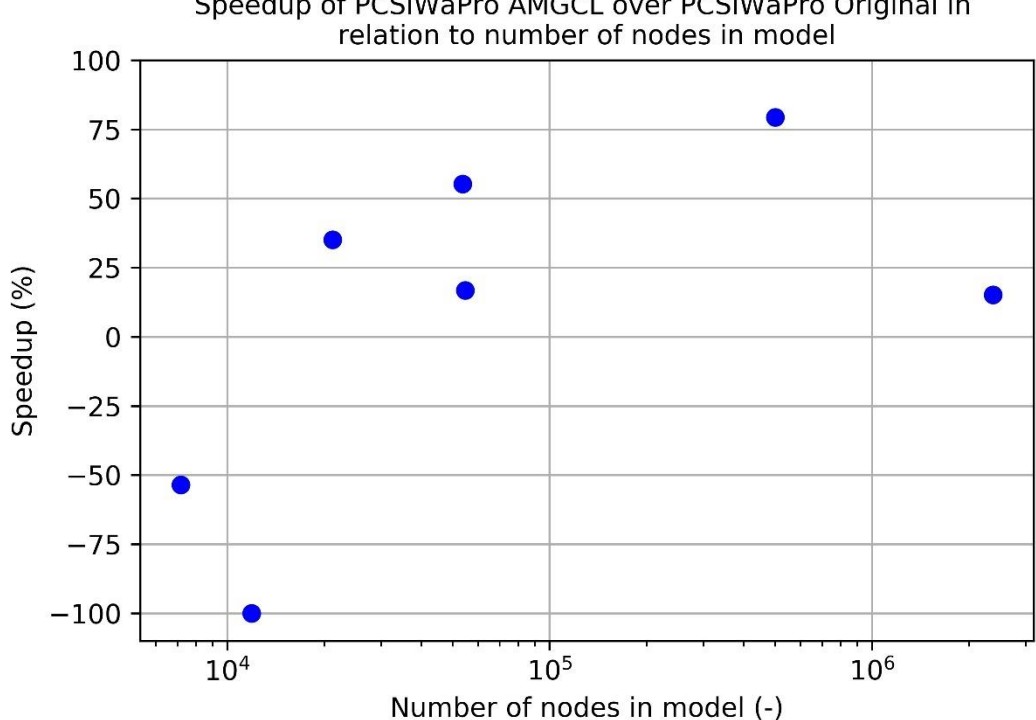

**Figure 3.** Comparison of the speedup achieved with PCSiWaPro AMGCL over PCSiWaPro Original in relation to the number of nodes in the model. One sees that PCSiWaPro AMGCL outperforms PCSiWaPro Original for models with 20,000 or more nodes.

During the simulation of Model 4 however, PCSiWaPro Original called the solver almost twice as often as PCSiWaPro AMGCL. One learns from Figure 4, that PCSiWaPro Original calls the solver more frequently in the second half of the simulation period than PCSiWaPro AMGCL. Thus, PCSiWaPro AMGCL achieves convergence easier than PCSiWaPro Original in the second half of the simulation period in Model 4.

The columns considering the number of solver calls in Table 2 tell, that PCSiWaPro AMGCL has problems with convergence in comparison with PCSiWaPro Original in Model 7 since PCSiWaPro AMGCL needs 14% more solver calls than PCSiWaPro Original. This also shows, that the average time spent with the solution of a given equation system arising from Model 7 is shorter in PCSiWaPro AMGCL than in PCSiWaPro Original, since PCSiWaPro AMGCL computes faster in general in Model 7 while making more calls to the solver than PCSiWaPro Original.

Therefore, models 4 and 7 show, that there is no single best solver: whereas in Model 4 PCSiWaPro AMGCL could achieve a speedup in comparison with PCSiWaPro Original due to its solver, in Model 7 the AMGCL solver achieves convergence with more solver calls than PCSiWaPro Original and thereby limits the speedup.

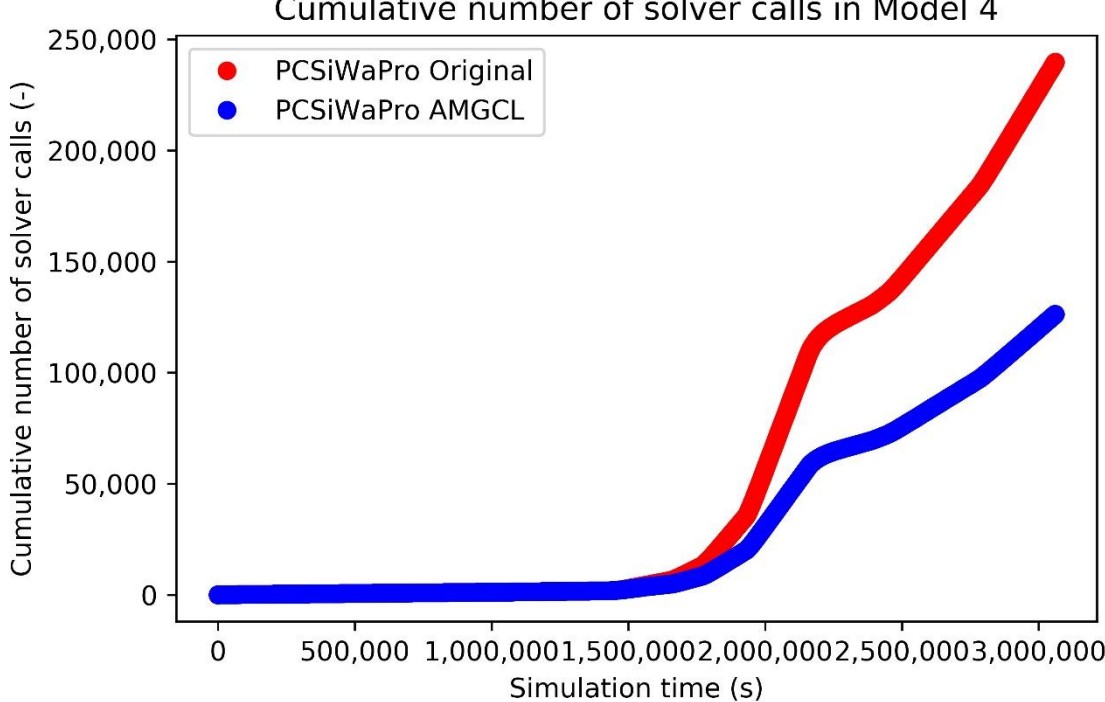

**Figure 4.** Comparison of the cumulative number of solver calls in Model 4. In the second half of the simulation period, PCSiWaPro AMGCL (blue) calls the solver less frequent than PCSiWaPro Original (red). This tells, that PCSiWaPro Original has greater problems with convergence than PCSiWaPro AMGCL.

*6.2. Comparison of the $R^2$-Values between the Pressure Heads Computed with PCSiWaPro Original and PCSiWaPro AMGCL*

Table 3 shows, that PCSiWaPro Original and PCSiWaPro AMGCL basically compute the same pressure heads. This means, that the speedup documented in Table 2 was not bought with a worsening of the quality of the computational results.

**Table 3.** Overview over the $R^2$-statistics between the pressure heads computed with PCSiWaPro Original and PCSiWaPro AMGCL.

| Model | Minimal $R^2$ | Maximal $R^2$ | Mean $R^2$ | Median $R^2$ |
|:-----:|:-------------:|:-------------:|:----------:|:------------:|
| - | - | - | - | - |
| 1 | 1.0 | 1.0 | 1.0 | 1.0 |
| 2 | 1.0 | 1.0 | 1.0 | 1.0 |
| 3 | 1.0 | 1.0 | 1.0 | 1.0 |
| 4 | 1.0 | 1.0 | 1.0 | 1.0 |
| 5 | 1.0 | 1.0 | 1.0 | 1.0 |
| 6 | 0.9998 | 1.0 | 1.0 | 1.0 |
| 7 | 1.0 | 1.0 | 1.0 | 1.0 |

*6.3. Comparison of the RCMBE Computed with PCSiWaPro Original and PCSiWaPro AMGCL*

In Table 4, a comparison of the RCMBE computed with PCSiWaPro Original and PCSiWaPro AMGCL is given. As one can see from Table 4, the models were tuned quite well, which can be read from the columns that represent the RCMBE statistics for PCSiWaPro Original, because this code was used during the tuning of the models.

**Table 4.** Comparison of the RCMBE computed with PCSiWaPro Original and PCSiWaPro AMGCL.

| Model | Relative Cumulative Mass Balance Error | | | | | | | |
|---|---|---|---|---|---|---|---|---|
| | % | | | | | | | |
| - | PCSiWaPro Original | | | | PCSiWaPro AMGCL | | | |
| | Minimal | Maximal | Mean | Median | Minimal | Maximal | Mean | Median |
| 1 | 0.0006 | 0.1621 | 0.0322 | 0.031 | 0.0004 | 0.7444 | 0.0164 | 0.013 |
| 2 | 0.0001 | 0.0043 | 0.0015 | 0.0013 | 0.0 | 0.0123 | 0.0027 | 0.0022 |
| 3 | 0.3390 | 0.70 | 0.4189 | 0.3983 | 0.4113 | 0.8826 | 0.6628 | 0.6623 |
| 4 | 0.0 | 0.0451 | 0.0037 | 0.0028 | 0.0004 | 0.2962 | 0.0192 | 0.0131 |
| 5 | 0.0116 | 0.9663 | 0.5696 | 0.6197 | 0.0055 | 1.8195 | 1.0265 | 1.1373 |
| 6 | 0.0 | 0.0373 | 0.0226 | 0.0201 | 0.0 | 0.2589 | 0.2097 | 0.223 |
| 7 | 0.0008 | 0.1787 | 0.0674 | 0.0602 | 0.0003 | 0.3137 | 0.1332 | 0.155 |

According to Table 4, only Model 5 needs to be investigated, since the RCMBE computed with PCSiWaPro AMGCL surpasses the threshold of 1%. In all the other models, this threshold is not violated by the computational results computed with PCSiWaPro AMGCL.

Figure 5 shows the RCMBE computed with PCSiWaPro Original (red) and with PCSiWaPro AMGCL (blue) in Model 5. In the first quarter of the simulation period, PCSiWaPro AMGCL violates the rule of keeping the RCMBE below 1%.

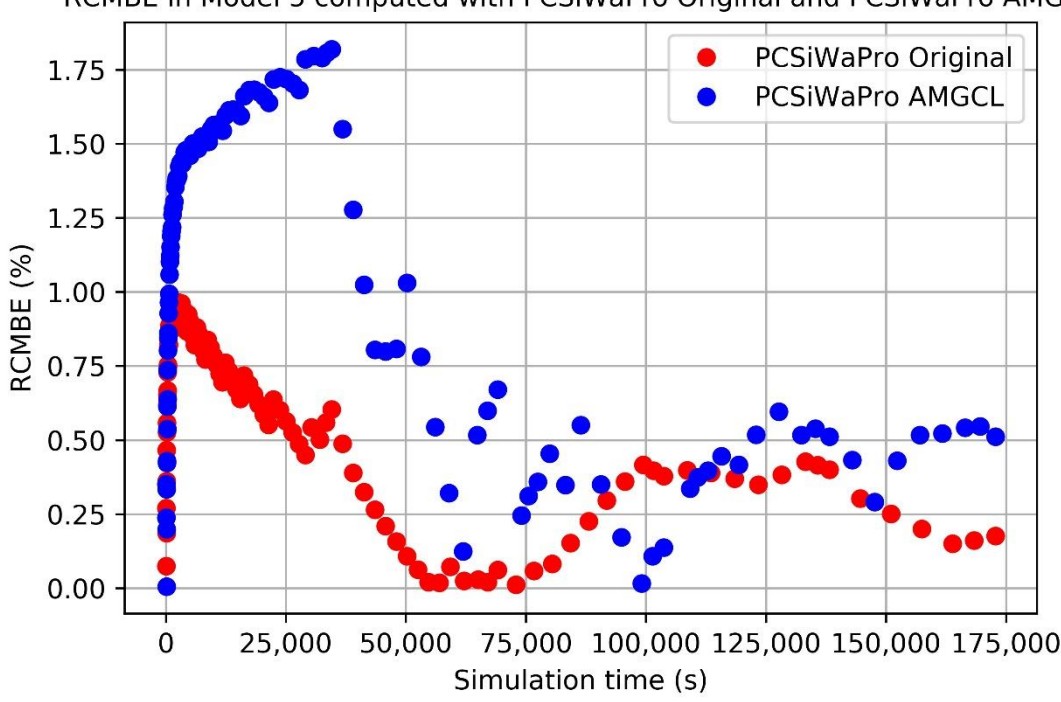

**Figure 5.** The RCMBE in Model 5 computed with PCSiWaPro Original (red) and PCSiWaPro AMGCL (blue). The results computed with PCSiWaPro Original do not violate the aim of a RCMBE of at most 1%. However, the results computed with PCSiWaPro AMGCL violate this rule in the first quarter of the simulation period.

Figure 6 shows the relationship between the computed influx and the computed RCMBE in both programs PCSiWaPro Original (red) and PCSiWaPro AMGCL (blue) for the first 10% of the simulation period in Model 5. PCSiWaPro solves Equation (1) for the pressure heads, therefore, since the boundary condition that creates the influx is a constant pressure head boundary condition, the computed pressure heads near this boundary condition must be very similar in both programs, otherwise the computed

influx would not be identical in both programs. Therefore, the difference in the RCMBE computed with PCSiWaPro Original and PCSiWaPro AMGCL can only be explained by different computational results for the pressure heads of nodes that are not near the constant pressure head boundary condition. The existence of such nodes can be seen in the $R^2$-statistics in Table 3.

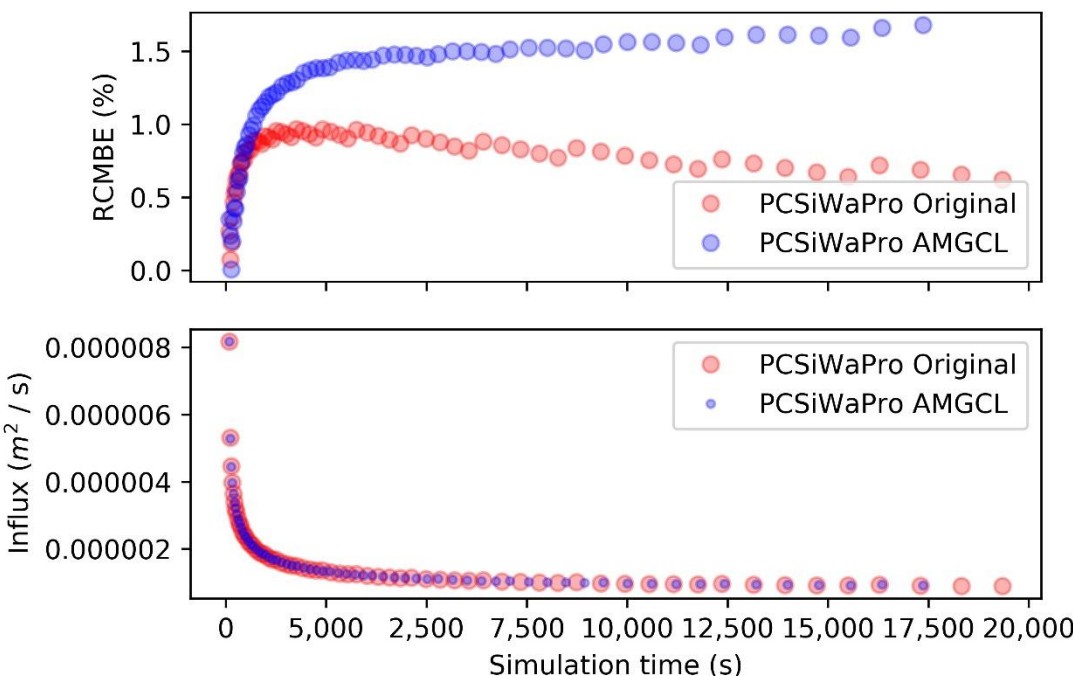

**Figure 6.** The relationship between the RCMBE computed with PCSiWaPro Original (red) and PCSiWaPro AMGCL (blue) and the computed influx for the first 10% of the simulation period in Model 5. Since the computed influx is identical in both programs, the difference in the mass balance must be explained by differently computed pressure heads.

This comparison of RCMBE shows that PCSiWaPro AMGCL performs slightly worse than PCSiWaPro Original.

## 7. Conclusions

The comparisons above show, that PCSiWaPro AMGCL computes results that are of similar quality as the results computed with PCSiWaPro Original while being faster for larger models (20,000+ nodes). In all cases considered in this paper, the $R^2$-values between the pressure heads computed with PCSiWaPro Original and PCSiWaPro AMGCL were in most models 1, and always greater than 0.999. Yet in one case, PCSiWaPro AMGCL computed results that violate against the rule to keep the RCMBE under 1%.

Also, for smaller models (<20,000 nodes), PCSiWaPro AMGCL computes slower than PCSiWaPro Original. Therefore, for larger models, PCSiWaPro Original may be replaced with PCSiWaPro AMGCL in order to speed up the computation.

The comparison of the number of calls of the solver showed, that the slower computational speed of PCSiWaPro AMGCL in comparison with PCSiWaPro Original in smaller models can only be explained by the numerical complexity of computing the algebraic Multigrid preconditioner. Therefore, for smaller models, the ILU-preconditioned Conjugate Gradient algorithm is a more efficient choice than BICGSTAB with algebraic Multigrid preconditioning.

This study only considered seepage flow that is modelled by Equation (1), but since BICCGSTAB can also deal with non-symmetric matrices [28], BICGSTAB preconditioned with an algebraic Multigrid

with Smoothed Aggregation and SPAI0 relaxation may be tested on transport problems in the vadose zone.

Another direction for future research is to evaluate whether further tuning of the models can increase the quality of the computational results computed with PCSiWaPro AMGCL while still being faster than PCSiWaPro Original.

Since PCSiWaPro uses the numerical discretization of the Richards' equation that is also used in SWMS and HYDRUS, the findings of this study should also hold for SWMS and HYDRUS.

However, one finding of this study is the poor speedup caused by AMGCL with the 3-dimensional Model 7. The speedup of 15% is too little to be of practical relevance. Therefore, the performance of PCSiWaPro AMGCL has to be further evaluated for larger models. Unless more promising results are produced, we can only recommend to apply AMGCL in the simulation of small and medium scale problems (20,000 to 500,000 nodes).

**Author Contributions:** Conceptualization, R.P.; methodology, R.P.; software R.P. and R.B.; validation, R.P. and R.B.; formal analysis, R.P. and R.B.; investigation, R.P. and R.B., resources, R.P. and R.B.; data curation, R.P. and R.B.; write-original draft preparation, R.P. and R.B.; writing-review and editing, R.P. and R.B.; visualization, R.P. and R.B. All authors have read and agreed to the published version of the manuscript.

**Funding:** This work was funded by the Saxon Development Bank (grant number 100301390), the European Social Funds of the European Union, and the German Research Foundation (grant number LI 727/24-1). Open Access Funding by the Publication Fund of the TU Dresden

**Acknowledgments:** We thank Denis Demidov for writing, maintaining and publishing of AMGCL, the solver library that was used in this study. We also want to express our gratitude to him due to his eagerness to answer our questions. Also, we would like to thank Falk Händel for providing Model 6. Furthermore, we thank Peter-Wolfgang Gräber and Rudolf Liedl for fruitful discussions about numerical mathematics.

**Conflicts of Interest:** The authors declare no conflict of interest. The funders had no role in the design of the study; in the collection, analyses, or interpretation of data; in the writing of the manuscript, or in the decision to publish the results.

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
