# Peer review of "Speeding up the Computation of the Transient Richards’ Equation with AMGCL"

_water, doi:10.3390/w12010286_

Round 1

Reviewer 1 Report

This paper provides a small numerical modeling study that uses a new C++ library to solve the groundwater flow equation in the vadose zone (i.e. Richard's equation). The results are very interesting, and show that numerical simulations can be performed at a higher speed than previous methods, at least for the numerical model that is used. However, I have several concerns that must be addressed before I can recommend publication.

The authors did not develop the new simulation software library (AMGCL), but rather used one that was already developed. The only advance made is to apply the library to a single modeling code, and use several small-scale simulations to demonstrate an incremental speed-up in the total simulation period. The scientific novelty of this is not clear. To be useful to readers and groundwater modelers in general, the library AMGCL should be applied to codes that are more often used: FEFLOW, HydroGeoSphere, and Hydrus, for example. And it should be applied at a larger scale, to show that indeed the Richard's equation can be a viable modeling tool at regional scales. Otherwise, we are still at the problem of not being able to implement the Richard's equation at scales often required for water management and contaminant transport.

How can AMGCL be employed with the other more commonly used groundwater modeling codes?

The comparison is only applied to one model (that is not often used by groundwater modelers); the comparison only happens at small spatial scales; therefore, the usefulness to the modeling community is not justified or apparent. A median increase in speed of 25-50% is not enough to make Richard's equation useful at regional scales. 

Please apply the library to a large spatial scale, and with more commonly used groundwater models. Then I would be happy to review the paper.

Reviewer 2 Report

The paper “Speeding up the computation of the transient Richards’ equation with AMGCL” focus on solving the Richards’ equations using free sAMGCL library. Authors’ report reduction of the computational time of up to 79% as compared to standard methods without losing accuracy. The paper presents computational testsof different sizes and conclude that the acceleration is strongly linked with the size of the system. Encouraging conclusion says that the greatest speedup was

achieved for the largest model.

Nevertheless, in opinion of the reviewer there are some major issues. It is questionable if the paper really is suited for the Water journal, because the paper deals with a comparative study of chosen solvers and the Richardson equation can be seen only as an example for this comparison. There is no novelty in the paper and any flow or water related problem is addressed.

It seems that this paper should be submitted to a journal dealing with numerical metods.

1. Introduction:

lines 53 -54:

“The Finite Volume Method is considered in theoretical works [23, 24], but not in readily

available software products.”

I do not agree with this stetmant. The Richardson equation can be (relatively easily) implemented in OpenFOAM CFD toolbox which uses FVM. Out of many examples of using OpenFOAM to implement new models see e.g.:

Bozza et.al. Cryogenics, Volume 80, Part 3, December 2016, https://doi.org/10.1016/j.cryogenics.2016.04.007

2. Method:

There is only one short mention of the methodology adopted:

”AMGCL was integrated into PCSiWaPro to solve equation systems arising from Finite Element

models of equation (1)”

but how were the individual terms of equations discretized ? Which were implicit which were explicit ? How any other researcher implement and check presented results without this knowledge ?

Reviewer 3 Report

The present manuscript is to discuss how AMGCL would reduce the computing time of Richards’ equation using six synthetic models. It is more like a technical paper rather than a research paper. As a technical paper, it would be valuable to adopt AMGCL in the unsaturated zone modeling. As a research paper, it needs more verification of its effectiveness using a real case application. The manuscript reads well overall, but needs a better structure and more descriptions. Below are the major comments that need to be addressed.

The overall review of numerical methods in Introduction is well done. However, it is too lengthy to have it all in the introduction. I would like to suggest to cut the introduction to Line 45 and have overall goals or major subjects to tackle at the end. Line 46 to the rest can be a separate section for numerical analysis in general for hydrological modeling. Also, I would like to recommend to concise the review mainly for Richards’ equation modeling. It is way too long for a manuscript. Line 198-200 finally introduces AMGCL, but they are too abrupt to continue for 2. Method. Although AMGCL is explained more in 2. Method, a more introductory description is needed for AMGCL. Line 216 shows the adaptation of PCSiWaPro, but its justification is not enough. Who did Authors choose only this single model over all others available? Better and more justification is needed. Line 267. The subtitle 3. Materials needs to have more specific subtitle such as Synthetic Modeling or Synthetic Modeling Description, etc. Figure 1 shows the six synthetic models with different colors, but they are simply confusing. What does the black color do? Are they inactive nodes? The figures need more detailed symbols or descriptions like boundary conditions, etc. Although they are explained in the model description, they should be identified in the figure. Results and Table 2. The negative speed up of Model 1 and 2 might have caused by overestimation? The models look too simple for your AMGCL integrated approach. The early peak of PCSiWaPro AMGCL in Figure 3 would be the same reason. May need more discussion. Where is Conclusion?

Reviewer 4 Report

Manuscript entitled "Speeding up the computation of the transient Richards’ equation with AMGCL" presents interesting study concerning application of Richards equation for flow in unsaturated zone and its efficient solving. Considering merit of this subject I can suggest major revision, but at the same time I see significant limits to consider this manuscript as a original scientific article due to the following reasons:

I do not agree that Richards equation represents the most accurate mathematical description of seepage flow in the vadose zone. It is only the most used approximation relating to the huge nonlinearities due to its theorethical drawbacks. This study does not include any methodological novelty related to the solving of the Richards equation. This study is interesting  because it shows application of algebraic multigrid preconditioner for BiGSTAB Krylov solver of system of equations arising from the Richards equation. However, I ask myself and Editor is it enough for original scientific article. This can be important finding in the context of the some other wider original scientific contribution. I suggest that authors solve at least one 3-D benchmark problem with more than ten million unknown head values that show robustness of the presented methodology and possible application to Richards heavy 3-D examples. I hope that in that case this free library can serve as a strong tool for some future unsolved problems.

Round 2

Reviewer 1 Report

The authors have done a nice job addressing all reviewer comments. I am still not convinced that a 15% reduction in simulation speed is deemed a novelty in the field. But it is an incremental result that can be published.

Reviewer 2 Report

In opinion of the reviewer the manuscript was significantly improved and can be published in the current form.

Reviewer 3 Report

First of all, I would like to commend the authors’ efforts to fully address the previous review comments in the revision. Although it would be still more like a technical paper rather than a research paper, it reads better and will attract some interest in modeling communities. Below are some minor comments.

The abstract needs more specific conclusions, especially in the last two sentences. For example, saying ‘... the speedup is linked to the size of the equation system…’ is too generic for an abstract. The speedup was measured or observed with models larger than 20,000 nodes? The new figure 1 is a lot better than the previous one. I would like to suggest more explanations of each color in the figure caption. It would help readers understand which color indicates what boundary conditions, etc. ‘Discussion’ better not go with ‘Conclusion.’ It looks like the additional parts of ‘Results’ are more for ‘Discussion,’ so either having it as a separate section or combining with ‘Results.’

Reviewer 4 Report

I think that revised manuscript is improved and now ready for publication.
